# Quantifying the Molecular Properties of the Elk Chronic Wasting Disease Agent with Mass Spectrometry

**DOI:** 10.3390/pathogens13111008

**Published:** 2024-11-16

**Authors:** Christopher J. Silva, Melissa L. Erickson-Beltran, Eric D. Cassmann, Justin J. Greenlee

**Affiliations:** 1Produce Safety and Microbiology Research Unit, Western Regional Research Center, Agricultural Research Service, United States Department of Agriculture, Albany, CA 94710, USA; melissa.erickson@usda.gov; 2Department of Veterinary Pathology, College of Veterinary Medicine, Iowa State University, Patterson Hall, 1800 Christensen Drive, Ames, IA 50011, USA; cassmann@iastate.edu; 3Virus and Prion Research Unit, National Animal Disease Center, Agricultural Research Service, United States Department of Agriculture, Ames, IA 50010, USA; justin.greenlee@usda.gov

**Keywords:** prion, transmissible spongiform encephalopathy, TSE, chronic wasting disease, CWD, elk, methionine, methionine sulfoxide, mass spectrometry, multiple reaction monitoring, recombinant prion protein, rPrP

## Abstract

Chronic wasting disease (CWD) is a prion disease afflicting wild and farmed elk. CWD prions (PrP^Sc^) are infectious protein conformations that replicate by inducing a natively expressed prion protein (PrP^C^) to refold into the prion conformation. Mass spectrometry was used to study the prions resulting from a previously described experimental inoculation of MM132, ML132, and LL132 elk with a common CWD inoculum. Chymotryptic digestion times and instrument parameters were optimized to yield a set of six peptides, TNMK, MLGSAMSRPL, LLGSAMSRPL, ENMYR, MMER, and VVEQMCITQYQR. These peptides were used to quantify the amount, the M132 and L132 polymorphic composition, and the extent of methionine oxidation of elk PrP^Sc^. The amount (ng/g brain tissue) of PrP^Sc^ present in each sample was determined to be: MM132 (5.4 × 10^2^ ± 7 × 10^1^), ML132 (3.3 × 10^2^ ± 6 × 10^1^ and 3.6 × 10^2^ ± 3 × 10^1^) and LL132 (0.7 × 10^2^ ± 1 × 10^1^, 0.2 × 10^2^ ± 0.2 × 10^1^, and 0.2 × 10^2^ ± 0.5 × 10^1^). The proportion of L132 polymorphism in ML132 (heterozygous) PrP^Sc^ from CWD-infected elk was determined to be 43% ± 2% or 36% ± 3%. Methionine oxidation was detected and quantified for the M132 and L132 polymorphisms in the samples. In this way, mass spectrometry can be used to characterize prion strains at a molecular level.

## 1. Introduction

Chronic wasting disease (CWD) is a prion disease of wild and farmed cervids that is endemic to North America (Appendix A) [1]. Unlike other prion diseases, CWD spreads readily among wild and farmed cervids through natural animal interactions and from contaminated environments [2,3]. It was first described in Rocky Mountain elk in 1982 [4]. CWD was brought to South Korea by imported CWD-infected Rocky Mountain elk [5,6,7]. More recently, CWD was found to have emerged in Norwegian, Swedish, and Finnish cervids (reindeer, red deer, and moose) independently of North American CWD (Appendix A) [8,9,10,11,12,13,14]. The Scandinavian strains of CWD are distinct from North American strains of CWD [9,15,16]. This illustrates the need to detect new CWD strains.

Prions (PrP^Sc^) are infectious proteins that propagate an infection by inducing a native protein (PrP^C^; native cellular prion protein) to refold into the prion conformation [17,18]. Their propagation is dependent upon the ability of PrP^C^ to refold into the PrP^Sc^ conformation [19,20]. PrP^C^ polymorphisms influence the progression of CWD in cervids [21]. Rocky Mountain elk possess a polymorphism (methionine (M) or leucine (L)) at position 132. Epidemiological studies showed that CWD-infected MM132 homozygous elk were overrepresented in wild and farmed elk populations [22]. The CWD incubation period of experimentally infected MM132 Rocky Mountain elk is the shortest; ML132 heterozygotes show a longer incubation period, and LL132 display a still longer one (Appendix A) [23,24]. When Rocky Mountain elk (MM132, ML132, or LL132) are experimentally infected with CWD, a different strain of CWD is found in the LL132 elk but not in the MM132 or ML132 animals, even though the same inoculum was used to infect all three elk genotypes [24,25,26].

Mass spectrometry is a flexible multiplex means of analyzing prions. The multiple reaction monitoring (MRM) method (Appendix A) has been used to quantify prions at the attomole (10^−18^ mole) range [27]. It has been used to quantify the composition of polymorphic variants in the PrP^Sc^ of scrapie infected sheep and CWD-infected deer [28,29,30]. It has been used to study the surface exposure of methionines in prions [31]. Although MRM has been used to study prions in other species, it is easily adapted to determining the properties of elk prions [30].

We used mass spectrometry to analyze the PrP^Sc^ from three polymorphic elk (MM132, ML132, and LL132) experimentally infected with CWD prions. Mass spectrometry was used to quantify the amount of PrP^Sc^ in three genotypes of CWD-infected elk, the proportion of each polymorphism (M132 and L132) in CWD-infected ML132 elk, and the extent of oxidation of each methionine in PrP^Sc^, as reported herein.

## 2. Materials and Methods

### 2.1. Chemicals

LC/MS grade acetonitrile, dithiothreitol, and water were sourced from Fisher Scientific (Pittsburgh, PA, USA); chymotrypsin (alpha; 3× crystallized zymogen) from Worthington Biochemical Corporation (Lakewood, NJ); and recombinant trypsin and all other reagents from Sigma-Aldrich (St. Louis, MO, USA). ^15^NH_4_Cl (99.7% isotopically pure) was purchased from Cambridge Isotope Laboratories, Inc. (Tewksbury, MA, USA).

Synthetic peptides were used to optimize the instrument parameters and were sourced from Elim Biopharmaceuticals (Hayward, CA, USA). Mass spectrometry was used to verify the peptide sequences, and all were of high (>95%) chemical purity, based on LC/UV analysis. Enzymatic digestion (trypsin, chymotrypsin, or both) of uniformly ^15^N-labeled rPrP was used to generate the necessary uniformly ^15^N-labeled peptides. Mass spectrometry-based analysis showed ^15^N label incorporation into the uniformly ^15^N-labeled PrP internal standard to be an estimated 99.7%.

### 2.2. Samples and Sample Preparation

Six samples of frozen brain tissue (cerebrum) were obtained from a previously published experiment (Table 1) [23,25]. The prions were enriched using a modified Bolton et al. procedure [32,33]. In particular, a 10% sarkosyl (sodium N-(dodecanoyl)-N-methylglycinate), 9.5 mM NaPO4 pH 8.5 solution was added to each sample to make a 10% (*w*/*v*) brain homogenate. Each sample was homogenized with an Omni homogenizer using a disposable probe to preclude cross contamination. After homogenization, each sample was clarified by centrifugation (16,000× *g*; 18 min). One mL of each of the clarified homogenates was separately diluted with 2 mL of a 10% sarkosyl solution and mixed. The resulting solutions were added to separate ultracentrifuge tubes and underlaid with a 20% sucrose solution. Each tube was sealed, placed in a rotor (Beckman 70.1 Ti), and then ultracentrifuged (145,000× *g* (r-average); 75 min; Beckman L8 80M). Each supernatant was removed and discarded with appropriate disposal. The resulting pellets were separately resuspended in 150 µL of an 8 M guanidine hydrochloride solution (GdnCl) and allowed to stand for 24 h to inactivate the prions [34]. The inactivated prions were removed from the BSL2 laboratory to a BSL1 laboratory.

A tabular summary of the previously published animal properties is included in the Appendix A.

### 2.3. Reduction and Alkylation of rPrP and Inactivated CWD Prions

Fresh solutions of dithiothreitol (DTT; 1M in water) and iodoacetamide (IA, 500 mM in water) were prepared. Samples of rPrP were dissolved in a 6M GdnCl solution and then processed. Samples of inactivated prions were processed in the 6M GdnCl used to inactivate them.

Each sample was buffered with 25 mM Tris-HCl pH 8.0 and reduced with 25 mM DTT for 30 min at 50 °C, with 5 min of sonication at 0 and 15 min. The solution was cooled to room temperature (25 °C) and IA was added to 75 mM; the reaction was allowed to proceed in the dark (RT; 45 min). After the IA reaction was completed, DTT was added to 25 mM to quench the reaction. 1.3 mL of cold (−20 °C) methanol was added to 200 µL of the quenched solution to precipitate the reduced and alkylated proteins and stored in a −20 °C freezer. After an hour, the solution was centrifuged (−11 °C; 20,000× *g*; 20 min) to pellet the precipitated proteins. The supernatant was discarded, and the pellet was resuspended in 0.5 mL of cold (−20 °C) 85% aqueous methanol and then centrifuged (−11 °C; 20,000× *g*; 20 min) again. The supernatant was discarded, the pellet dried, and then stored at −80 °C until ready for enzymatic digestion.

### 2.4. Preparation of Recombinant PrP

The cloning of the mature (25–233; Appendix A) 132M (GenBank # EU082289.1) and 132L (GenBank # EU082286.1) Rocky Mountain elk prion proteins into the pET11a vector and then to BL21 cells has been described previously [30,33,35]. Glycerol stocks of these clones were prepared by adding 700 µL of an overnight Luria broth (LB) culture supplemented with carbenicillin (100 µg/mL) to 300 µL of sterile 50% aqueous glycerol. The stocks were stored at −80 °C until a small sample was streaked out onto LB agar plates supplemented with carbenicillin (100 µg/mL) and incubated overnight in a 37 °C incubator. A single colony was used to inoculate the minimal medium cultures.

^14^N-M9 (natural abundance) minimal medium (84.5 mM Na_2_HPO_4_, 44.4 mM KH_2_PO_4_, 17.1 mM NaCl, 37.4 mM ^14^NH_4_Cl, 2 mM MgSO_4_, 0.1 mM CaCl_2_, 33.2 µM thiamine, 22.2 mM glucose, and trace metals) was prepared. A single colony was used to inoculate a 250 mL baffled flask containing 25 mL of ^14^N-M9 medium supplemented with carbenicillin (100 µg/mL) and allowed to grow overnight in a shaker incubator (37 °C; 250 rpm). A 1 L baffled flask containing 150 mL of ^14^N-M9 medium supplemented with carbenicillin (100 µg/mL) was inoculated with 10 mL of the overnight culture and allowed to grow in a shaker incubator (37 °C; 250 rpm) until the OD_600_ of the culture was between 0.4 and 0.6. Once the culture had grown sufficiently, it was supplemented with 150 µL of a 1M sterile filtered aqueous solution of isopropyl β-D-thiogalactopyranoside (IPTG) and allowed to grow for an additional 4 h in the shaker incubator (37 °C; 250 rpm). The cells were pelleted by centrifugation (10,000× *g*; 5 min). The supernatant was discarded, and the pellet was retained for further processing.

^15^N-M9 minimal medium (84.5 mM Na_2_HPO_4_, 44.4 mM KH_2_PO_4_, 17.1 mM NaCl, 37.4 mM ^15^NH_4_Cl, 2 mM MgSO_4_, 0.1 mM CaCl_2_, 33.2 µM thiamine, 22.2 mM glucose, and trace metals) was prepared. A single colony was used to inoculate a 250 mL baffled flask containing 25 mL of ^15^N-M9 medium supplemented with carbenicillin (100 µg/mL) and allowed to grow overnight in a shaker incubator (37 °C; 250 rpm). One mL of the overnight culture was removed and pelleted by centrifugation (3000× *g*; 3 min). The supernatant was discarded, and the pellet was resuspended in sterile water. A fresh 250 mL baffled flask containing 25 mL of ^15^N-M9 medium supplemented with carbenicillin (100 µg/mL) was inoculated with 10 µL of the sterile water suspension and allowed to grow overnight (37 °C; 250 rpm). A 1 L baffled flask containing 150 mL of ^15^N-M9 medium supplemented with carbenicillin (100 µg/mL) was inoculated with 10 mL of the overnight culture and allowed to grow in a shaker incubator (37 °C; 250 rpm) until the OD_600_ of the culture was between 0.4 and 0.6. Once the culture had grown sufficiently, it was supplemented with 150 µL of a 1M sterile filtered aqueous solution of isopropyl β-D-thiogalactopyranoside (IPTG) and allowed to grow for an additional 4 h in the shaker incubator (37 °C; 250 rpm). The cells were pelleted by centrifugation (10,000× *g*; 5 min). The supernatant was discarded, and the pellet was retained for further processing. The pellet contained uniformly ^15^N-labeled cells.

Inclusion bodies were isolated from the natural abundance or ^15^N-labeled pellets using standard molecular biology techniques. The inclusion bodies were resuspended in 6 M GndCl, 100 mM NaPO4 pH 8.0, and 10 mM Tris-HCl pH 8.0, and purified by immobilized metal affinity chromatography (IMAC) using previously described procedures. The IMAC fraction containing the purified natural abundance recombinant PrP (rPrP) or ^15^N-labeled rPrP was dialyzed against ammonium acetate (100 mM, pH 4.5, overnight), followed by another 2 h dialysis against fresh ammonium acetate (50 mM, pH 4.5). The retentate was apportioned by 0.5 mL aliquots into microcentrifuge tubes and lyophilized overnight. The lyophilized samples of natural abundance or uniformly ^15^N-labeled rPrP were stored at −80 °C until needed.

### 2.5. Enzymatic Digestion

A stock solution of chymotrypsin (1 mg/mL chymotrypsin) in storage buffer (1 mM HCl, 2 mM CaCl_2_) was prepared per the manufacturer’s recommendations. A stock solution of trypsin (1 mg/mL) in storage buffer (1 mM HCl) was prepared per the manufacturer’s recommendations. Both solutions were stored in a 4 °C fridge, as recommended by the manufacturers, until needed.

Three alternate methods of digesting the pellets were attempted. In the first method, pellets containing reduced and alkylated rPrP or inactivated prions were resuspended in 20 µL of a solution of 0.01% β-octylglucopyranoside (BOG), 1 pmol/μL methionine, and 8% acetonitrile, and then sonicated (50 °C; 5 min). Then, 79 µL of digestion buffer (25 mM ammonium bicarbonate (ABC), 0.01% β-octylglucopyranoside (BOG), 1 pmol/μL methionine, and 8% acetonitrile) was added and sonicated again (50 °C; 5 min). One µL (=1 µg) of the trypsin solution was added to each sample and incubated at 37 °C overnight. 50 µL of the trypsin digest was quenched by the addition of 1.25 µL of a 10% aqueous formic acid solution. The solution was centrifuged (14,000× *g*; 12 min) through a twice-washed (0.5 mL water) 10,000 MWCO centrifuge filter. The filtrate was stored at −80 °C until ready for analysis.

In the second method, the remaining 50 µL of the trypsin digest was digested with chymotrypsin (500 ng chymotrypsin, 2 mM CaCl_2_ per reaction; 30 °C) for various times (5, 10, 15, 30, 45, or 60), depending on the experiment. After the timed digestion was complete, the chymotryptic digestion was quenched by the addition of 1.25 µL of a 10% aqueous formic acid solution and centrifuged through a 10,000 MWCO, twice-washed centrifugal filter. The filtrate was stored at −80 °C until ready for analysis.

In the third method, pellets containing reduced and alkylated rPrP or inactivated prions were redissolved in 20 µL of a solution of 0.01% β-octylglucopyranoside (BOG), 1 pmol/μL methionine, and 8% acetonitrile, and then sonicated (50 °C; 5 min). Then, 79 µL of digestion buffer (25 mM ammonium bicarbonate (ABC), 0.01% β-octylglucopyranoside (BOG), 1 pmol/μL methionine, and 8% acetonitrile) was added and sonicated again (50 °C; 5 min). Chymotryptic digestion was effected by adding 500 ng of chymotrypsin solution and CaCl_2_ to 2 mM, and digesting for 15 min at 30 °C. After the digestion, 1.25 µL of a 10% aqueous formic acid solution was added to quench the digestion. Each digest was centrifuged (14,000× *g*; 12 min) through a twice-washed (0.5 mL water) 10,000 MWCO centrifuge filter. The filtrate was stored at −80 °C until ready for analysis.

### 2.6. Peptide Optimization

Relevant peptides were obtained from commercial vendors. The optimal multiple reaction monitoring (MRM) parameters were obtained by a previously described method [27]. The MRM parameters for previously optimized ^14^N tryptic peptides and ^14^N chymotryptic peptides are summarized in Appendix A, respectively. The analogous MRM parameters for ^15^N tryptic peptides and ^15^N chymotryptic peptides are summarized in Appendix A, respectively.

### 2.7. Mass Spectrometry

An Applied Biosystems Tempo nanoflow LC system (Applied Biosystems Tempo nanoflow LC system (AB Sciex LLC; Framingham, MA, USA)) with an autosampler, a column switching device, and a nanoflow solvent delivery system was used to deliver the mobile phase. Six µL of each digest was loaded onto a trapping cartridge (C-18; Acclaim PepMap100, 5 µm, 100 Å, 300 µm ID × 5 mm (Thermo Scientific Dionex, Sunnyvale, CA, USA)). The cartridge was washed with an acetic acid/acetonitrile/heptafluorobutyric acid/water solution (0.5/1/0.02/98.48%) for 3–5 min (5 µL/min). The flow was switched, and the bound peptides eluted onto a reversed-phase column (Vydac Everest (HiChrom, Leicestershire, UK) 238EV5.07515, 75 µm ID × 150 mm). A binary gradient (A, 2% acetonitrile with 0.5% acetic acid in water; B, 99.5% acetonitrile with 0.5% acetic acid) was used to elute the samples.

A linear gradient (30 min; 300 nL/min) began with 95% A and ended with 10% A. The 10% A was held for 20 min and then returned to 95% A over 5 min and equilibrated at this composition for an additional 5 min. The eluant was continuously sprayed through a spray tip (Non coated; FS360-20- 10-N-20-C12, New Objective Inc., Woburn, MA, USA) using the Applied Biosystems Nanospray II electrospray assembly.

Quantitative mass spectrometry was performed using an Applied Biosystems (AB Sciex LLC; Framingham, MA, USA) model 4000 Q-Trap instrument. The instrument was operated in MRM mode. It was set to alternate between detecting the natural abundance (^14^N) analyte peptides and the analogous ^15^N-labeled internal standard peptides. The parameters for each peptide and the corresponding ^15^N-labeled analog were empirically determined using previously described methods. These parameters are summarized in Appendix A. The IntelliQuan quantification algorithm (Analyst 1.6.3 software; AB Sciex) was used to quantitate the natural abundance and ^15^N-labeled peptides.

### 2.8. Preparing ^15^N-Labeled Internal Standards

Uniformly labeled ^15^N-labeled elk rPrP (132M or 132L) was reduced and alkylated. The reduced and alkylated protein was digested overnight with trypsin, quenched with formic acid, and filtered to yield a set of uniformly ^15^N-labeled tryptic peptides that were used as internal standards for the analysis of the natural abundance peptides derived from the tryptic digestion of the prion protein samples.

Uniformly ^15^N-labeled elk (132M or 132L) chymotryptic peptides were obtained by two means. For the sake of visual clarity, the **M** or **L** polymorphism at position 132 is bolded and underlined. An unquenched portion of the overnight trypsin was digested with chymotrypsin for an additional 15 min, quenched with formic acid, and filtered to yield uniformly ^15^N-labeled **M**LGSAMSR and **L**LGSAMSR that were used as internal standards to analyze the natural abundance peptides derived from the tryptic/chymotryptic digestion of prion protein samples. In the second approach, uniformly labeled ^15^N-labeled elk rPrP (132M or 132L) was reduced and alkylated, digested with chymotrypsin alone for 15 min, quenched with formic acid, and filtered to yield uniformly ^15^N-labeled **M**LGSAMSRPL and **L**LGSAMSRPL. The peptides were used as internal standards to analyze the natural abundance of peptides derived from the chymotryptic digestion of the prion protein samples.

### 2.9. Preparation of Calibration Curves for Quantitation of PrP^Sc^

Samples of uniformly ^15^N-labeled internal standard tryptic peptides were prepared. Dilutions of solutions (10, 5, 2, 1, 0.5, 0.1, or 0.05 femtomole/injection) of the commercially acquired ^14^N-VVEQMCITQYQR analyte peptide were prepared. A fixed amount of the uniformly ^15^N-labeled internal standard tryptic peptides was added to each dilution. The areas of the MRM signals for the ^14^N-VVEQMCITQYQR and ^15^N-VVEQMCITQYQR peptides were determined for each dilution.

### 2.10. Determining the Proportion of 132M and 132L Polymorphisms in a Sample

The samples of PrP (M132 or L132) were digested with chymotrypsin. The integrated areas of the MRM signals for **M**LGSAMSRPL or **L**LGSAMSRPL and RYPNQVY were determined. The ratio of the areas for **M**LGSAMSRPL or **L**LGSAMSRPL to the RYPNQVY peptide were determined for MM132 and LL132 samples, respectively. These empirically determined area ratios were used to normalize the **M**LGSAMSRPL and **L**LGSAMSRPL to the RYPNQVY peptide in samples from heterozygous (ML132 samples) elk.

### 2.11. Determining the Amount of Oxidation in a Peptide

The optimized MRM parameters were determined previously for the four tryptic peptides (TNMK, ENMYR, MMER, and VVEQMCITQYQR) common to the 132M and 132L elk PrP polymorphisms and two chymotryptic peptides (**M**LGSAMSR and **L**LGSAMSR, or **M**LGSAMSRPL and **L**LGSAMSRPL). The optimized transitions for these and other peptides are summarized in Appendix A. The integrated areas of the unoxidized and oxidized peptides were determined. The percentage oxidation was reported as the area of the oxidized or unoxidized peptides divided by the sum of the areas of the oxidized and unoxidized peptides.

### 2.12. Analysis of Sample Data

The IntelliQuan quantification algorithm (Analyst 1.6.3 software; AB Sciex) was used to quantitate the natural abundance (^14^N) and ^15^N-labeled peptides. The software reported the integrated areas of the MRM signals for the optimized peptides in the method. The areas of methionine-containing peptides used in calculations are the sum of the MRM signals from the unoxidized methionine-containing peptide, oxidized methionine-containing (methionine sulfoxide) peptide, and artifactual ESI oxidation of the methionine-containing peptide. The areas of peptides without methionine correspond to the reported area of the MRM signal for the methionine-free peptide.

Statistical and regression analysis was performed using Excel (2022).

## 3. Results

### 3.1. Optimizing Chymotryptic Digestion

The peptides suitable for an MRM-based analysis and containing the 132M and 132L polymorphisms are contained in chymotryptic peptides. The **M**LGSAMSR and **L**LGSAMSR peptides are produced in low amounts in an overnight trypsin digestion of elk PrP. The same peptides are produced in much greater abundance by digesting the overnight trypsin digest with chymotrypsin. The **M**LGSAMSRPL and **L**LGSAMSRPL peptides are produced by digesting the sample with chymotrypsin alone. These three approaches yield peptides that can be used to quantify the amount of the M132 and L132 polymorphisms in a sample from a CWD-infected elk. The optimal time for the chymotrypsin digestion needed to produce the peptides **M**LGSAMSR and **L**LGSAMSR, or **M**LGSAMSRPL and **L**LGSAMSRPL, was restricted to 15 min after an overnight trypsin digestion or without any trypsin digestion, respectively.

### 3.2. Selecting Conditions to Minimize Matrix Effects

A modified method of Bolton et al. was used to isolate the PrP^Sc^, as it isolates more than 95% of the infectivity present in the sample [32]. Furthermore, our approach does not use proteinase K (PK) to remove non-prion proteins. This means that the tryptic or chymotryptic digestion of proteins other than PrP^Sc^ may interfere with the analysis. Overnight digestion with trypsin yielded the tryptic peptides TNMK, ENMYR, MMER, and VVEQMCITQYQR. There was minimal matrix background interference for these tryptic peptides (Appendix A). The other peptides needed for this analysis required the use of chymotrypsin.

A sample of MM132 and LL132 CWD-infected elk was digested overnight with trypsin, followed by 15 min with chymotrypsin, or 15 min with chymotrypsin without prior trypsin digestion. The MRM chromatograms from the overnight trypsin digest of the MM132 and LL132 CWD-infected elk showed a significant amount of matrix background for the transitions corresponding to the **M**LGSAMSR and the **L**LGSAMSR peptides (Appendix A). When these overnight tryptic digests were further digested with chymotrypsin, the MRM chromatograms showed a greater signal for the **M**LGSAMSR peptide, but the **L**LGSAMSR peptide was obscured by a matrix contaminant (Appendix A). When the samples were digested with chymotrypsin alone, the matrix background for the transitions corresponding to the **M**LGSAMSRPL or **L**LGSAMSRPL peptides was minimal and the signal-to-noise ratio was greatly improved (Appendix A).

The samples were digested overnight with trypsin to yield a set of tryptic peptides that include methionines 112 (TN**M**K), 157 (EN**M**YR), 208 and 209 (**MM**ER), and 216 (VVEQ**M**CITQYQR). The peptides containing the M132 polymorphism and methionines 132 and 137 (**M**LGSA**M**SRPL), or the L132 polymorphism and methionine 137 (**L**LGSA**M**SRPL), are derived from a short chymotryptic digestion.

### 3.3. Determining the Percentage of M132 and L132 in a Sample

In addition to the **M**LGSAMSRPL and **L**LGSAMSRPL peptides, chymotryptic digestion yields the RYPNQVY peptide (positions 159–165). The **M**LGSAMSRPL and **L**LGSAMSRPL peptides are produced in proportion to the M132 and L132 polymorphisms. The RYPNQVY peptide is produced in proportion to the amount of PrP in the sample and not the amount of M132 or L132 polymorphism. The area ratios of the MRM signals of the **M**LGSAMSRPL peptide to that of the RYPNQVY peptide for the (MM132) Park10 sample and the internal standard were determined to be 3.1 ± 0.8. The area ratios of the MRM signals of the **L**LGSAMSRPL peptide to that of the RYPNQVY peptide for the three homozygous (LL132) Valley samples (1, 3, and 4) and the internal standard were determined to be 1.2 ± 0.2. These ratios relate the area of the MRM signals from the **M**LGSAMSRPL and **L**LGSAMSRPL peptides to that of a peptide common to both polymorphisms, RYPNQVY.

These ratios were used to normalize the observed areas of MRM signals from the **M**LGSAMSRPL and **L**LGSAMSRPL peptides to an equivalent area of MRM signal for the RYPNQVY peptide, for each polymorphism. These calculations were performed on the **M**LGSAMSRPL and **L**LGSAMSRPL peptides derived from the heterozygous elk (Park 11 and Park 7). The calculated area of the RYPNQVY peptide based on the **M**LGSAMSRPL and **L**LGSAMSRPL areas differed from the observed area of the RYPNQVY peptide by 12.8% and 11.0% for the Park 10 and Park 7 samples, respectively. The calculated RYPNQVY values were used to determine the percentage composition of the M132 and L132 polymorphisms in PrP^Sc^ in the six samples. These values are reported in Table 2.

### 3.4. A Calibration Curve to Quantitate the Amount of PrP^Sc^ in a Sample

The VVEQMCITQYQR tryptic peptide is common to both the M132 and L132 polymorphisms. It is well-suited to quantitate the amount of PrP^Sc^ in a sample from homozygous or heterozygous CWD-infected animals. The calculated areas of the MRM signals from each ^14^N-VVEQMCITQYQR dilution and the fixed amount of the added ^15^N-VVEQMCITQYQR internal standard peptide were used to prepare the calibration curve. The area ratios of the ^14^N-VVEQMCITQYQR dilution to the fixed amount of added ^15^N-VVEQMCITQYQR internal standard peptide were plotted vs. the amount of ^14^N-VVEQMCITQYQR peptide in each dilution. These data were graphed (Appendix A). Regression analysis showed the linear curve to have an excellent correlation coefficient (R^2^ = 0.9997) with high significance (*p* < 0.0001). This calibration curve permits the quantitation of the total amount of CWD PrP^Sc^ present in each sample, regardless of the polymorphic composition, since the VVEQMCITQYQR peptide is common to both the 132M and 132L polymorphic variants.

### 3.5. Oxidation Studies

Four tryptic peptides (TNMK, ENMYR, MMER, and VVEQMCITQYQR) and two chymotryptic peptides (**M**LGSAMSRPL and **L**LGSAMSRPL) contain the methionines present in the M132 or L132 elk PrP^Sc^. The methionines at positions 112, 157, 208, 209, and 216 are present in the TNMK, ENMYR, MMER, and VVEQMCITQYQR tryptic peptides, respectively. The methionines present at positions 132 and 137 in the M132 polymorphism and the methionine at position 137 in the L132 polymorphism are contained in the tryptic peptides **M**LGSAMSRPL or **L**LGSAMSRPL, respectively. Each peptide was analyzed using the previously optimized MRM parameters and chromatography conditions for the peptide [31]. The integrated areas of the MRM signals for each sample’s appropriate unoxidized and oxidized peptide were determined. These values were used to calculate the percentage of oxidized and unoxidized methionine in each peptide present in the sample. The results are summarized in Table 3.

### 3.6. Quantitation

The integrated areas of the MRM signals from the oxidized and unoxidized tryptic analyte peptide (VVEQMCITQYQR) were determined for each sample. The areas of the MRM signals for the unoxidized and oxidized peptide were combined to yield the total area of the analyte peptide for each sample. The previously prepared calibration curve (vide supra) was used to relate the experimentally determined areas of the MRM signals to the corresponding amount of the VVEQMCITQYQR in the sample. As this peptide is common to the M132 and L132 polymorphic variants of the elk prion protein and it is produced in proportion to the PrP in the sample, it provides a measure of the amount of PrP^Sc^ in the sample. These results are summarized in Table 4.

## 4. Discussion

The multiple reaction monitoring method (MRM), also known as selective reaction monitoring (SRM), is an established mass spectrometry method (Appendix A) that has been used to analyze prions. This approach detects the peptides resulting from the tryptic or chymotryptic digestion of a denatured prion and is a means of analyzing relevant peptide segments of the denatured protein. The MRM method filters masses, which means that samples do not need to be extensively purified before being analyzed. Despite filtering, the sample matrix contains molecules that interfere with the analysis of tryptic digests.

A modified version of the method of Bolton et al. was used to isolate the PrP^Sc^ [32,33]. The samples used in this study were from the cerebrum but are not matched in the sense that they are not all from the same portion of the cerebrum. The gray/white matter composition of the samples is unknown, which may skew the results, since spongiform lesions are more prominent in the gray matter for MM132 and ML132 prions compared to LL132, where spongiform are more prominent in white matter. Proteinase K (PK) is commonly used to digest PK sensitive proteins, leaving the PK resistant truncated form of the prion, PrP 27–30 [18]. Previous work showed that a small amount of PrP^C^ is isolated from elk samples using this method, but that amount does not constitute a significant contribution relative to the amount of PrP^Sc^ in a sample [35]. PK was not used in order to avoid the significant loss of PK-sensitive PrP^Sc^ that is associated with the PK treatment of elk PrP^Sc^ [35]. It also means that tryptic or chymotryptic digestion of proteins other than PrP^Sc^ may interfere with the analysis.

Digestion of the elk samples yields a set of tryptic peptides. Four of these peptides (TNMK, ENMYR, MMER, and VVEQMCITQYQR) contain one or two methionines each and are common to both the M132 and L132 polymorphic variants of elk PrP. No significant matrix interference was observed for these four peptides nor their oxidized analogs (Appendix A). The other two or one methionines are contained in the **M**LGSAMSR (132M) or **L**LGSAMSR (132L) peptides, respectively, which are produced in low amounts by trypsin digestion. Unfortunately, the matrix interference precluded using trypsin digestion to produce these peptides (Appendix A). Supplemental digestion with chymotrypsin increased the yield of these peptides, but it was not sufficient to overcome the matrix interference (Appendix A). Digesting elk rPrP (132M or 132L) with chymotrypsin yields a suitable peptide, **M**LGSAMSRPL (132M) or **L**LGSAMSRPL (132L), for each polymorphism. When CWD-infected elk samples were digested with chymotrypsin, no matrix interference with the MRM signals of these peptides was observed (Appendix A). Since no matrix interference was observed for these six peptides (TNMK, **M**LGSAMSRPL, **L**LGSAMRPL, ENMYR, MMER, and VVEQMCITQYQR), they can be used to quantify the amount of PrP^Sc^ in a sample, determine the polymorphic composition of PrP^Sc^, and calculate the extent of oxidation of each methionine in PrP^Sc^.

A calibration curve was prepared to quantify the amount of PrP^Sc^ in each sample (Appendix A). The VVEQMCITQYQR peptide is a tryptic digest product of either (M132 or L132) polymorphism, so it can be used to quantify the amount of PrP^Sc^ in homozygotes (MM132 or LL132) and heterozygotes (ML132). Since the samples are not processed with PK, the prions consist of PK-sensitive and PK-resistant PrP^Sc^. The amount of PrP^Sc^ present in the MM132 homozygote was approximately 1.5-fold greater than that found in the ML132 heterozygous animals and approximately 7- to 27-fold greater than that found in the LL132 homozygotes. These values are roughly comparable to empirical observations seen in the Western blot-based analysis [24].

The relative amounts of the three polymorphic variants were determined by the antigen-capture enzyme immunoassay (EIA) [25]. The EIA proportions are approximately 2- to 6-fold greater in MM132 compared to LL132 and approximately 1.3-fold greater in MM132 compared to ML132. The EIA estimates are based on tissue from the midbrain (obex). As noted previously, the tissue used in this study is from unmatched cerebral samples whose gray and white matter composition is not known. The EIA study used tissue from the brainstem (obex). Mass spectrometry-based quantitation accurately measures the amount of PrP^Sc^ in a sample. Unfortunately, these samples are probably not representative of the PrP^Sc^ distribution in the brainstem, hence the disparity between these results and those resulting from the EIA-based analysis.

The empirical relationship between the areas of the MRM signals of the chymotryptic peptides **M**LGSAMSRPL, **L**LGSAMSRPL, and RYPNQVY was established. The **M**LGSAMSRPL and **L**LGSAMSRPL signals were normalized to the RYPNQVY peptide and used to determine the composition of PrP^Sc^ in heterozygous animals. Most of the ML132 PrP^Sc^ was composed of the M132 (57% ± 2% or 64% ± 3%) polymorphism. Interestingly, a significant minority of ML132 PrP^Sc^ (43% ± 2% or 36% ± 3%) contained L132. The composition of PrP^Sc^ from white-tailed deer experimentally infected with CWD has been determined by mass spectrometry [30]. When GS96 (glycine (G) or serine (S) at position 96) heterozygous deer are experimentally infected with CWD, the composition of the resulting PrP^Sc^ is 75% ± 5% G and 25% ± 5% S, as measured by mass spectrometry [30]. Our results are consistent with those observed for experimentally infected deer, in the sense that the more resistant polymorphism (S96) is a lesser component of the mixture. In sheep naturally infected with scrapie, the more resistant polymorphism (Histidine (H) at position 154 or alanine (A) at position 136) predominated [28]. Mass spectrometry can be used to quantify the proportion of different (e.g., M132 and L132) polymorphisms present in PrP^Sc^ from heterozygous animals (e.g., ML132 elk).

Other researchers analyzed the migration patterns of MM132, ML132, and LL132 PrP^Sc^ by Western blot [23,24]. The Western blots of MM132 and ML132 PrP^Sc^ showed identical migration patterns that were different from the migration pattern of LL132 PrP^Sc^. The migration pattern of MM132 and ML132 PrP^Sc^ showed no evidence of the LL132 PrP^Sc^ migration pattern [24]. Our analysis showed that a significant (41% ± 4%) amount of the L132 is present in the ML132 PrP^Sc^. Despite the incorporation of L132, ML132 PrP^Sc^ has the same conformation as the MM132 PrP^Sc^, as determined by Western blot analysis. This suggests that L132 can adopt the MM132 PrP^Sc^ conformation.

The stability of the MM132, ML132, and LL132 PrP^Sc^ fibrils was determined by other researchers [25]. These results showed that the MM132 and ML132 PrP^Sc^ fibrils have similar stabilities. In contrast, the LL132 PrP^Sc^ fibrils are much more stable, as a higher concentration of guanidine hydrochloride is needed to denature them. Again, this evidence supports the observation that even though L132 is present in significant amounts in ML132 PrP^Sc^, it adopts the MM132 and ML132 PrP^Sc^ conformation.

Even though L132 can be induced to adopt the 132MM PrP^Sc^ conformation, it imparts some phenotypic differences onto ML132 PrP^Sc^. The incubation period of prions in MM132 elk is the shortest (23 months); LL132 (63 ± 2 months) is the longest; and, in between, ML132 (39 ± 1 months) PrP^Sc^ (Appendix A) [23]. The extension of the incubation period in ML132 elk is thought to be due to a 50% reduction in M132 substrate in heterozygous (ML132) animals [25]. Since the amount of L132 in ML132 PrP^Sc^ is ~40%, ML132 PrP^Sc^ propagation may be impeded both by the lack of the preferred M132 PrP^C^ substrate (50% of total PrP^C^) and the abundance of a poorer L132 PrP^C^ substrate (50% of total PrP^C^). If PrP^Sc^ amplification is impeded, then the smaller, more infectious, and more PK-sensitive multimers become more susceptible to native cellular protein recycling machinery. The result of this would be a longer incubation period and a lower amount of PrP^Sc^. When either MM132 or ML132 PrP^Sc^ is passaged through MM132 transgenic mice (*Tg12*), the resulting prions have comparable incubation times and fibril stabilities [26]. This further confirms that the L132 component of ML132 PrP^Sc^ has adopted the MM132 PrP^Sc^ conformation.

A protein’s methionines are susceptible to oxidation by reactive oxygen species. Mammalian cells express enzymes (methionine sulfoxide reductases) that reduce oxidized methionines in intracellular proteins to minimize the damage caused by reactive oxygen species [36,37]. The refolded PrP^C^ comprising prions contains reduced methionines. If the methionines on the surface of the prions outside of a cell or in dead cells are oxidized, then they cannot be reduced by intracellular methionine sulfoxide reductases. Methionines in the interior of a prion are less susceptible to oxidation. Oxidized methionines or unoxidized methionines remain so after a prion has been inactivated by denaturation. These retained covalent modifications can be used to infer the relative surface exposure of methionines in the prion conformation.

The methionine-containing peptides (TNMK, **M**LGSAMSRPL, **L**LGSAMSRPL, ENMYR, MMER, and VVEQMCITQYQR) were readily detectable and quantifiable. These peptides contain methionines (met) at positions 112, 132 and 137 (M132), 137 only (L132), 157, 208 and 209, and 216, respectively. In the elk samples, the TNMK (met112) and MMER (met208 and met209) peptides are largely unoxidized (≥90%) for all samples, which suggests they may have limited surface exposure in the prion conformation. In ML132 prions, **M**LGSAMSRPL and **L**LGSAMSRPL are also largely unoxidized (≥83%), suggesting comparable surface exposure. In LL132 prions, **L**LGSAMSRPL is more oxidized (60–84% unoxidized), but there is less of the peptide present in these samples, which generally results in greater levels of oxidation. In most samples, met157 (ENMYR peptide) is more oxidized than met112 and more so when the amount of peptide in the sample is lower. In most samples, met216 is largely unoxidized (>94%). The two samples in which met216 is more heavily oxidized (14 and 21%) are from samples with the lowest MRM signal intensity (Valley 3 and 4; 2.5 × 10^4^ ± 4 × 10^3^; Table 2).

Methionines can be oxidized in the prion conformation, or artifactually oxidized in the denatured prion, or as individual peptides. While these results hint at quantifiable conformation-dependent differences in methionine oxidation, it is also possible that the observed oxidation is artifactual, which makes the result difficult to interpret. Samples with lower amounts of PrP^Sc^, such as the three LL132 PrP^Sc^ samples, are proportionally more susceptible to artifactual oxidation than more concentrated samples [35,38,39]. A means of overcoming this limitation is to oxidize a sample with hydrogen peroxide, which does not significantly perturb the prion conformation, and compare the extent of methionine oxidation before and after chemical oxidation [31].

Mass spectrometry can be used to detect features of prions that are not detectable by other means. It can also be used to quantify prions and determine the polymorphic composition of PrP^Sc^ from prion-infected heterozygous animals. Devising alternative digests can be used to overcome matrix effects. A small set of methionine containing peptides can be used to quantify the proportion of oxidized methionine.

## Figures and Tables

**Table 1 pathogens-13-01008-t001:** A summary of the elk samples. The elk # corresponds to the animals described in a previous publication [25]. The elk IDs are used in this manuscript. * Sample not available; not analyzed by mass spectrometry.

Elk #	Elk ID	Genotype	Amount (g)
1	Park 10	M/M	1
2	---	---	---*
3	Park 11	L/M	1
4	Park 7	L/M	1
5	Valley 4	L/L	1
6	Valley 3	L/L	1
7	---	---	---*
8	Valley 1	L/L	1.3

**Table 2 pathogens-13-01008-t002:** The percentage of the L132 and M132 polymorphisms in CWD PrP^Sc^ present in the six CWD-infected elk samples. The animals, genotypes, and percentage (two samples per animal; 4 injections/animal) of the M132 and L132 in the samples are summarized.

Animal	Genotype	% M132	% L132
Park 10	M/M	100	0
Park 11	M/L	57 ± 2	43 ± 2
Park 7	M/L	64 ± 3	36 ± 3
Valley 1	L/L	0	100
Valley 3	L/L	0	100
Valley 4	L/L	0	100

**Table 3 pathogens-13-01008-t003:** A summary of the extent of oxidation of methionines 112, 132, 137, 157, 208, 209, and 216 as determined by quantifying the percentage of the unoxidized and oxidized forms of the TNMK (met112) and **M**LGSAMSRPL (met132 and met137) or LLGSAMSRPL (met 137), ENMYR (met157), MMER (met208 and met209), and VVEQMCITQYQR (met216) peptides (Appendix A) for each sample (n = 4). * Valley 1, 3, and 4 are homozygous (LL132); the **M**LGSAMSRPL peptide was not detected in the chymotryptic digests of these Valley (1, 3, and 4) samples. ^†^ Park 10 is homozygous (MM132); the **L**LGSAMSRPL peptide was not detected in the chymotryptic digests of the Park 10 sample. ND (not detected).

**Animal**	**Genotype**	**Signal Intensity**	**% Met112**	**% MetSO112**		
Park 10	M/M	7.7 × 10^5^ ± 7 × 10^4^	98.8 ± 0.3	1.2 ± 0.3		
Park 11	M/L	8.0 × 10^5^ ± 4 × 10^4^	97 ± 2	3 ± 2		
Park 7	M/L	9.8 × 10^5^ ± 7 × 10^4^	98.3 ± 0.9	1.7 ± 0.9		
Valley 1	L/L	2.4 × 10^5^ ± 1 × 10^4^	94.2 ± 0.8	5.8 ± 0.8		
Valley 3	L/L	0.9 × 10^5^ ± 1 × 10^4^	90 ± 2	10 ± 2		
Valley 4	L/L	2.4 × 10^5^ ± 3 × 10^4^	91 ± 2	9 ± 2		
**Animal**	**Genotype**	**Signal Intensity**	**% Met132 + Met137**	**% MetSO132 + Met137**	**% Met132 + MetSO137**	**% MetSO132 + MetSO137**
Park 10	M/M	2.3 × 10^6^ ± 4 × 10^4^	86.4 ± 0.8	5.0 ± 0.2	7.2 ± 0.3	1.5 ± 0.7
Park 11	M/L	3.1 × 10^5^ ± 1 × 10^4^	84.7 ± 0.6	5.0 ± 0.5	7.0 ± 0.6	3.3 ± 0.5
Park 7	M/L	1.4 × 10^6^ ± 2 × 10^5^	87 ± 1	5 ± 1	5.8 ± 0.5	2 ± 1
Valley 1	L/L	---*	---*	---*	---*	---*
Valley 3	L/L	---*	---*	---*	---*	---*
Valley 4	L/L	---*	---*	---*	---*	---*
**Animal**	**Genotype**	**Signal Intensity**	**% Met137**	**% MetSO137**		
Park 10	M/M	---^†^	---^†^	---^†^		
Park 11	M/L	1.0 × 10^5^ ± 2 × 10^3^	83 ± 1	17 ± 1		
Park 7	M/L	3.3 × 10^5^ ± 3 × 10^4^	87.2 ± 0.8	12.8 ± 0.8		
Valley 1	L/L	1.0 × 10^5^ ± 1 × 10^4^	84 ± 2	16 ± 2		
Valley 3	L/L	1.0 × 10^5^ ± 1 × 10^4^	71 ± 3	29 ± 3		
Valley 4	L/L	0.7 × 10^5^ ± 1 × 10^4^	60 ± 4	40 ± 4		
**Animal**	**Genotype**	**Signal Intensity**	**% Met157**	**% MetSO157**		
Park 10	M/M	6.1 × 10^6^ ± 1 × 10^5^	87.9 ± 0.8	12.1 ± 0.8		
Park 11	M/L	4.4 × 10^6^ ± 2 × 10^5^	82.2 ± 0.5	17.8 ± 0.5		
Park 7	M/L	6.1 × 10^6^ ± 1 × 10^5^	92.6 ± 0.3	7.4 ± 0.3		
Valley 1	L/L	4.6 × 10^5^ ± 5 × 10^4^	82 ± 2	18 ± 2		
Valley 3	L/L	3.1 × 10^5^ ± 1 × 10^4^	72 ± 4	28 ± 4		
Valley 4	L/L	3.8 × 10^5^ ± 3 × 10^4^	69 ± 1	31 ± 1		
**Animal**	**Genotype**	**Signal Intensity**	**% Met208 + Met209**	**% MetSO208 + Met209**	**% Met208 + MetSO209**	**% MetSO208 + MetSO209**
Park 10	M/M	8.5 × 10^6^ ± 7 × 10^4^	98.8 ± 0.3	0.1 ± 0.2	1.1 ± 0.2	ND
Park 11	M/L	6.6 × 10^6^ ± 2 × 10^5^	99 ± 0.2	ND	1 ± 0.2	ND
Park 7	M/L	7.7 × 10^6^ ± 8 × 10^4^	99 ± 0.3	0.5 ± 0.2	0.5 ± 0.1	ND
Valley 1	L/L	6.7 × 10^5^ ± 6 × 10^4^	92.7 ± 0.9	3.7 ± 0.8	3.5 ± 0.3	ND
Valley 3	L/L	8.9 × 10^5^ ± 5 × 10^4^	96.9 ± 2.1	1.9 ± 2.2	1.2 ± 0.2	ND
Valley 4	L/L	8.5 × 10^5^ ± 3 × 10^4^	98.8 ± 0.1	ND	1.2 ± 0.1	ND
**Animal**	**Genotype**	**Signal Intensity**	**% Met216**	**% MetSO216**		
Park 10	M/M	2.1 × 10^5^ ± 2 × 10^4^	95 ± 1	5 ± 1		
Park 11	M/L	2.1 × 10^5^ ± 3 × 10^4^	95 ± 1	5 ± 1		
Park 7	M/L	1.7 × 10^5^ ± 2 × 10^4^	94 ± 1	6 ± 1		
Valley 1	L/L	1.2 × 10^5^ ± 9 × 10^3^	94 ± 2	6 ± 2		
Valley 3	L/L	2.5 × 10^4^ ± 3 × 10^3^	86 ± 4	14 ± 4		
Valley 4	L/L	2.5 × 10^4^ ± 4 × 10^3^	79 ± 3	21 ± 3		

**Table 4 pathogens-13-01008-t004:** The amount of CWD PrP^Sc^ present in the six CWD-infected elk. The animals, their genotypes, and the amounts (ng PrP^Sc^/g brain tissue; n = 4) are listed.

Animal	Genotype	Amount of PrP^Sc^
Park 10	M/M	5.4 × 10^2^ ± 7 × 10^1^
Park 11	M/L	3.3 × 10^2^ ± 6 × 10^1^
Park 7	M/L	3.6 × 10^2^ ± 3 × 10^1^
Valley 1	L/L	0.7 × 10^2^ ± 1 × 10^1^
Valley 3	L/L	0.2 × 10^2^ ± 0.2 × 10^1^
Valley 4	L/L	0.2 × 10^2^ ± 0.5 × 10^1^

## Data Availability

Data are contained within the article and Appendix A.

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
