# Peer review of "Quantifying the Molecular Properties of the Elk Chronic Wasting Disease Agent with Mass Spectrometry"

_pathogens, 2024, doi:10.3390/pathogens13111008_

Round 1

Reviewer 1 Report

Comments and Suggestions for Authors

This was a really interesting paper about the application of MRM in prion diseases to circumvent extensive purification steps for quantitative mass spectrometry. Mass spectrometry will become more important to identify post-translational and conformational features of prions, that are not readily identified by other means.

Unfortunately, my understanding of the highly specialised science of and experimental designs used in mass spectrometry to provide a more in-depth analysis of this manuscript.

Author Response

The reviewer made the following comment:  

This was a really interesting paper about the application of MRM in prion diseases to circumvent extensive purification steps for quantitative mass spectrometry. Mass spectrometry will become more important to identify post-translational and conformational features of prions, that are not readily identified by other means.

Unfortunately, my understanding of the highly specialised science of and experimental designs used in mass spectrometry to provide a more in-depth analysis of this manuscript.

I agree with the reviewer.

Reviewer 2 Report

Comments and Suggestions for Authors

This is a well written technical report, showing how to use modern mass spectrometry to obtain quantitative information of prion isolates. 

Author Response

The reviewer made the following comment:  This is a well written technical report, showing how to use modern mass spectrometry to obtain quantitative information of prion isolates. 

I agree with the reviewer.

Reviewer 3 Report

Comments and Suggestions for Authors

The manuscript, “Quantifying the molecular properties of the elk chronic wasting disease agent with mass spectrometry” by Silva et al., provides valuable information on the composition and abundance of specific molecules in the composition of the CWD prions from Elk with different genetic backgrounds. This is an important study as it provides inference into the compatibility of different prion primary structures to be incorporated into growing CWD prions. I am supportive of publication, after some major revisions are undertaken to sharpen the manuscript’s focus and, where possible, clarify ambiguities or properly limit the interpretations by describing the uncertainties associated with the experimental design. I hope that these comments are helpful to the authors.

The introduction does not well present the nature and scope of the problem investigated and it does not indicate why the elk chronic wasting disease prions require characterization and why mass spectrometry is the tool of choice. Much of the brief review is not pertinent or could be better described by referencing. From reading the introduction, I do not know what the objective is. Please reduce the use of scientific names for the species in the introduction. It is distracting. Please correct the scientific name for Elk (Cervus canadensis). Condense the first paragraph to Cervidae and cite a review? Some of the introductory material seems to be unrelated. For example, details on Fallow deer prp are introduced, but never again mentioned. Why is this important to introduce? Much of lines 48-55 could be referenced with the previous sentence without additional explication. Some of the discussion is introductory. The authors might consider if some of that material could be moved to sharpen the focus of the introduction and help the reader understand why the experiments were performed. Most importantly, provide a few more sentences summarizing previous findings and describing the importance of LL, ML and MM Elk PrP and how that affects the disease in Elk.

A few more details on the source of materials used for these studies would be beneficial. Moore et al., 2018 describe the full experiment and therein the samples are referred to as 1-8. How do the descriptors (Park 10, 11,7 and Valley 1,3,4 correspond to animals 1-8 from the Moore et al. Study? Apparently, there is another MM and LL sample that was not analyzed? Please help the reader out by not forcing them to dig for key details.

Line 86, Please describe in greater detail what is meant by “brain tissue.” What part of the brain was used in each sample? This is important as prion abundance may vary by brain region and so an alternative explanation might be that the brain regions were not matched.

Line 104, Is it 6M or 8M GDN. Please clarify.

The VVEQMCITQYQR peptide considered suitable to quantitate the amount of PrPSc in the samples. The RYPNQVY peptide is considered suitable for determining the proportion of M or L prions. Presumably any or all the common peptides could have been used as indicated in the abstract. Why were the specific peptides chosen for the different quantifications and why not use a generalized assessment based upon data derived from all the common peptides? Perhaps readers are missing something here as we may not be experts on mass spectrometry.

The amount of 132L incorporated into the PrPCWD is appreciable in the heterozygotes. I think this is unexpected. How should this be understood in the context of the compatibility of these prions? Why is disease extended in the M/L animals if the 132L primary structures are readily incorporated?

What experiments did the authors perform to determine that PrPCWD from 132MM, 132ML and 132LL was equally purified by the ultracentrifugation method chosen. How do they know that prion aggregates composed of 132LL are not selectively lost in the pellet of the clarification step.  Perhaps this data is available, in which case it should be included. Alternatively, the discussion could recognize that the purification approach could have been biased. I do not accept as definitive the sentence that the Bolton method isolates more than 95% of the infectivity as a Bolton prep was not used in this manuscript and that approach is describing the purification of hamster prions, which are expected to have different biochemical properties from elk prions. Do LL derived prions have different biochemical properties from MM prions from ML prions and if so how confident can we be on their unbiased purification?

The authors describe how methione oxidation quantification can provide information about prion conformation and yet no such inference appears to be drawn from the data. Instead, the results “hint” at a potential conformation. Would the authors please share that hint with the reader and appropriately limit the interpretation? Is there a reason that tables 2-7 cannot be merged? Why is the % Met and MetSO reported? Is one not just the subtraction from 100 the other? In table 6, one column seems to be mislabeled % MetSO + 208 Met209.

 524-528, I do not understand the discussion related to peroxide. This does not seem to be relevant. Please delete or add details to indicate how this is related to the present work.

Author Response

Reviewer comment:

The manuscript, “Quantifying the molecular properties of the elk chronic wasting disease agent with mass spectrometry” by Silva et al., provides valuable information on the composition and abundance of specific molecules in the composition of the CWD prions from Elk with different genetic backgrounds. This is an important study as it provides inference into the compatibility of different prion primary structures to be incorporated into growing CWD prions. I am supportive of publication, after some major revisions are undertaken to sharpen the manuscript’s focus and, where possible, clarify ambiguities or properly limit the interpretations by describing the uncertainties associated with the experimental design. I hope that these comments are helpful to the authors.

Response:  We appreciate the Reviewer’s careful reading of the manuscript and have endeavored to incorporate their suggestions into the manuscript.  The introduction has been revised to focus on elk CWD [lines 41-73].

Reviewer Comment:

The introduction does not well present the nature and scope of the problem investigated and it does not indicate why the elk chronic wasting disease prions require characterization and why mass spectrometry is the tool of choice. Much of the brief review is not pertinent or could be better described by referencing. From reading the introduction, I do not know what the objective is. Please reduce the use of scientific names for the species in the introduction. It is distracting. Please correct the scientific name for Elk (Cervus canadensis). Condense the first paragraph to Cervidae and cite a review? Some of the introductory material seems to be unrelated. For example, details on Fallow deer prp are introduced, but never again mentioned. Why is this important to introduce? Much of lines 48-55 could be referenced with the previous sentence without additional explication. Some of the discussion is introductory. The authors might consider if some of that material could be moved to sharpen the focus of the introduction and help the reader understand why the experiments were performed. Most importantly, provide a few more sentences summarizing previous findings and describing the importance of LL, ML and MM Elk PrP and how that affects the disease in Elk.

Response:  The introduction has been revised to conform with the reviewer’s suggestions [lines 41-73].

Reviewer Comment:

A few more details on the source of materials used for these studies would be beneficial. Moore et al., 2018 describe the full experiment and therein the samples are referred to as 1-8. How do the descriptors (Park 10, 11,7 and Valley 1,3,4 correspond to animals 1-8 from the Moore et al. Study? Apparently, there is another MM and LL sample that was not analyzed? Please help the reader out by not forcing them to dig for key details.

Response:  A table has been added to the experimental section which translates 1-8 numbering to Park 10, 11, 7, Valley 1, 3, 4 designations [Lines93-94, 107-114; Table 1].  It also includes the amount of sample available for analysis.  In addition, a table summarizing the previous results (incubation periods, IHC results, etc.) has been included in the supporting information (Table S1).  

Reviewer comment:

Line 86, Please describe in greater detail what is meant by “brain tissue.” What part of the brain was used in each sample? This is important as prion abundance may vary by brain region and so an alternative explanation might be that the brain regions were not matched.

Response:  The brain tissue samples are from the cerebrum [Lines93-94, 107-114; Table 1].  They are comparable samples, in the sense they are from the cerebrum.  The samples are not matched in terms of the amount of grey or white matter present in the sample or the specific portion of the cerebrum.  These were the last available samples.  There were no alternates.

Reviewer comment:

Line 104, Is it 6M or 8M GDN. Please clarify.

Response:  We prepare an 8M guanidine hydrochloride (GdnCl) solution that can be added to a prion-containing solution in a 3-fold excess to yield a final concentration of 6M GdnCl.  The pellet has a volume, so we add the 8M GdnCl solution to ensure that the final concentration is at least 6M GdnCl.

Reviewer comment:

The VVEQMCITQYQR peptide considered suitable to quantitate the amount of PrPSc in the samples. The RYPNQVY peptide is considered suitable for determining the proportion of M or L prions. Presumably any or all the common peptides could have been used as indicated in the abstract. Why were the specific peptides chosen for the different quantifications and why not use a generalized assessment based upon data derived from all the common peptides? Perhaps readers are missing something here as we may not be experts on mass spectrometry.

Response:  Different peptides were selected for different purposes.  For example, the VVEQMCITQYQR, RYPNQVY, and YPGQGSPGGNR peptides have been used to quantify the amount of PrP in a sample in published studies.[1-3]  Other researchers have used different tryptic peptides to quantify domains of human PrP.[4]

VVEQMCITQYQR was selected because it has good MRM properties, is detectable in the attomole range, and is present in PK-treated or PK-untreated samples.   The YPGQGSPGGNR (and PGGGWNTGGSR) peptides are not present in PrP 27-30, so they would only be suitable for quantifying full-length PrP.   The asparagine (N) of the GENFTETDIK peptide is variably glycosylated, which precludes its selection as an analyte peptide.   Other peptides (PLIHFGNDYEDR and YPNQVYYR) are the product of an R/P cleavage which has lower, variable efficiency and thus makes them unsuitable for quantitation.  The C-terminal portion of the ESEAYYQR peptide is cleaved during shedding, which makes it unsuitable for quantitation.  This is why VVEQMCITQYQR is considered suitable for quantifying the amount of PrP in a sample that is digested with trypsin.[1]

Other researchers have used RYPNQVY to quantify the amount of PrP in a sample that was digested with chymotrypsin.[3]  We also found this peptide to be suitable for quantifying the amount of PrP in a chymotryptic digest.

VVEQMCITQYQR is present in a tryptic digestion, but not in a chymotryptic digestion.  RYPNQVY is present in a chymotprytic digestion, but not a tryptic digestion.  Hence the need to use the two different peptides.

References

[1]         Onisko, B.; Dynin, I.; Requena, J. R.; Silva, C. J.; Erickson, M.; Carter, J. M. Mass spectrometric detection of attomole amounts of the prion protein by nanoLC/MS/MS. J. Am. Soc. Mass Spectrom. 2007, 18, 1070-1079.

[2]         Douma, M. D.; Kerr, G. M.; Brown, R. S.; Keller, B. O.; Oleschuk, R. D. Mass spectrometric detection of proteins in non-aqueous media — The case of prion proteins in biodiesel. Can. J. Chem. 2008, 86, 774-781.

[3]         Sturm, R.; Sheynkman, G.; Booth, C.; Smith, L. M.; Pedersen, J. A.; Li, L. Absolute quantification of prion protein [90-231] using stable isotope-labeled chymotryptic peptide standards in a LC-MRM AQUA workflow. J. Am. Soc. Mass Spectrom. 2012, 23, 1522-1533.

[4]         Minikel, E. V.; Kuhn, E.; Cocco, A. R.; Vallabh, S. M.; Hartigan, C. R.; Reidenbach, A. G.; Safar, J. G.; Raymond, G. J.; McCarthy, M. D.; O'Keefe, R.; Llorens, F.; Zerr, I.; Capellari, S.; Parchi, P.; Schreiber, S. L.; Carr, S. A. Domain-specific Quantification of Prion Protein in Cerebrospinal Fluid by Targeted Mass Spectrometry. Mol. Cell. Proteomics 2019, 18, 2388-2400.

Reviewer comment:

The amount of 132L incorporated into the PrPCWD is appreciable in the heterozygotes. I think this is unexpected. How should this be understood in the context of the compatibility of these prions? Why is disease extended in the M/L animals if the 132L primary structures are readily incorporated?

Response:  Our previous work with deer may provide some insight to this question.[1] In white tailed deer, the G/S polymorphism at position 96 influences the incubation time of CWD.  The PrPSc isolated from experimentally (orally dosed) infected heterozygous (GS96) white tailed deer was composed of ~ 75 % G and 25 % S.[1]  It is clear that the GS96 PrPSc is composed of both polymorphisms. The incubation period of the GS96 white tailed deer is longer than that of the GG96 white tailed deer.  In sheep naturally infected with scrapie, HQ171 heterozygotes show 60% H and 40 % Q in their PrPSc; AV136 heterozygotes show 70 % A and 30 % V.[2]  Since these sheep are naturally infected, the inoculum is not known nor is the incubation time.   These results indicate that PrPSc from heterozygous animals often contains both polymorphisms in non-equal amounts.

132L comprises ~ 40% of the PrPSc from M/L132 elk.  These results suggest that M132 PrPC is more readily refolded into ML132 PrPSc and that L132 PrPC is less readily refolded into ML132 PrPSc.  This implies that the replication of the ML132 PrPSc is impeded both by the lack (50 % of total PrPC) of the preferred M132 PrPC substrate and the abundance (50 % of total PrPC) of a poorer L132 PrPC substrate.  If PrPSc amplification is impeded, then the smaller, more infectious, and more PK sensitive multimers become more susceptible to native cellular protein recycling machinery.  The result of this would be a longer incubation period and a lower amount of PrPSc

References

[1]         Silva, C. J.; Erickson-Beltran, M. L.; Duque Velasquez, C.; Aiken, J. M.; McKenzie, D. A General Mass Spectrometry-Based Method of Quantitating Prion Polymorphisms from Heterozygous Chronic Wasting Disease-Infected Cervids. Anal. Chem. 2020, 92, 1276-1284.

[2]         Silva, C. J.; Erickson-Beltran, M. L.; Hui, C.; Badiola, J. J.; Nicholson, E. M.; Requena, J. R.; Bolea, R. Quantitating PrP Polymorphisms Present in Prions from Heterozygous Scrapie-Infected Sheep. Anal. Chem. 2017, 89, 854-861.

Reviewer comment:

What experiments did the authors perform to determine that PrPCWD from 132MM, 132ML and 132LL was equally purified by the ultracentrifugation method chosen. How do they know that prion aggregates composed of 132LL are not selectively lost in the pellet of the clarification step.  Perhaps this data is available, in which case it should be included. Alternatively, the discussion could recognize that the purification approach could have been biased. I do not accept as definitive the sentence that the Bolton method isolates more than 95% of the infectivity as a Bolton prep was not used in this manuscript and that approach is describing the purification of hamster prions, which are expected to have different biochemical properties from elk prions. Do LL derived prions have different biochemical properties from MM prions from ML prions and if so how confident can we be on their unbiased purification?

Response:  The authors performed no experiments to determine if MM132, ML132, and LL132 PrPSc were equally isolated by the ultracentrifugation method.  It is possible that the LL132 PrPSc is pelleted to a greater extent than the MM132 and ML132 PrPSc.  We discarded the low-speed pellets, so we have no way of determining if LL132 PrPSc is disproportionately retained in the low-speed pellet.  This has been included in the discussion.  It should be noted that both proteinase K sensitive and resistant hamster PrPSc have been isolated using the modified Bolton et al. method.[1,2]  The statement of 95% purity is an accurate reporting of amount of infectivity present in the supernatant after the low speed spin.[3]  

LL132 PrPSc fibrils are more stable than ML132 and MM132 PrPSc.  The spongiform lesions in brain tissue infected with ML132 and ML132 prions are more apparent in grey matter than in white matter.[4]  For LL132 prions, spongiform lesions are more apparent in the white matter than grey matter.10  The cerebrum samples are not matched for grey and white matter composition. If a sample containing LL132 PrPSc is composed mostly of grey matter it may contain less LL132 PrPSc than a sample containing more white matter.  The section has been rewritten [Lines 451-456, 487-496].  Mass spectrometry can be used to quantify the amount of PrPSc in a sample.  Unfortunately, these samples are not representative of the brainstem (obex) samples that were analyzed by EIA.[4]

References

[1]         Pastrana, M. A.; Sajnani, G.; Onisko, B.; Castilla, J.; Morales, R.; Soto, C.; Requena, J. R. Isolation and characterization of a proteinase K-sensitive PrPSc fraction. Biochemistry 2006, 45, 15710-15717.

[2]         Sajnani, G.; Silva, C. J.; Ramos, A.; Pastrana, M. A.; Onisko, B. C.; Erickson, M. L.; Antaki, E. M.; Dynin, I.; Vazquez-Fernandez, E.; Sigurdson, C. J.; Carter, J. M.; Requena, J. R. PK-sensitive PrP is infectious and shares basic structural features with PK-resistant PrP. PLoS Pathog. 2012, 8, e1002547.

[3]         Bolton, D. C.; Rudelli, R. D.; Currie, J. R.; Bendheim, P. E. Copurification of Sp33-37 and scrapie agent from hamster brain prior to detectable histopathology and clinical disease. J. Gen. Virol. 1991, 72 [ Pt 12], 2905-2913.

[4]         Moore, S. J.; Vrentas, C. E.; Hwang, S.; West Greenlee, M. H.; Nicholson, E. M.; Greenlee, J. J. Pathologic and biochemical characterization of PrP[Sc] from elk with PRNP polymorphisms at codon 132 after experimental infection with the chronic wasting disease agent. BMC Vet. Res. 2018, 14, 80.

 Reviewer comment:

The authors describe how methione oxidation quantification can provide information about prion conformation and yet no such inference appears to be drawn from the data. Instead, the results “hint” at a potential conformation. Would the authors please share that hint with the reader and appropriately limit the interpretation? Is there a reason that tables 2-7 cannot be merged? Why is the % Met and MetSO reported? Is one not just the subtraction from 100 the other? In table 6, one column seems to be mislabeled % MetSO + 208 Met209.

Response:  Tables 2, 3, 4, 5, 6, and 7 (old numbering) have been merged into Table 3 (new numbering) [Lines 410-424].  The reviewer is correct in observing that % Met = 100% - MetSO %.  Both are included to avoid any confusion for a less careful reader.  The mislabeled column in Table 6 has been corrected.  I apologize for the oversight.

I have rewritten the section to clarify “the hint”, the limitations of our analysis of methionine-containing peptides, and suggested an approach to determine the surface exposure of a prion’s methionine by quantifying the extent of methionine oxidation after reaction with hydrogen peroxide.

 Reviewer comment:

 524-528, I do not understand the discussion related to peroxide. This does not seem to be relevant. Please delete or add details to indicate how this is related to the present work.

Response:  The section has been rewritten and details added to make it relevant [Lines 538-567].

Round 2

Reviewer 3 Report

Comments and Suggestions for Authors

I have no further concerns.